# Effects of Mild Excitotoxic Stimulus on Mitochondria Ca^2+^ Handling in Hippocampal Cultures of a Mouse Model of Alzheimer’s Disease

**DOI:** 10.3390/cells10082046

**Published:** 2021-08-10

**Authors:** Giulia Rigotto, Lorena Zentilin, Tullio Pozzan, Emy Basso

**Affiliations:** 1Department of Biomedical Sciences, University of Padua, 35131 Padua, Italy; giulia.rigotto@iov.veneto.it (G.R.); Tullio.Pozzan@unipd.it (T.P.); 2International Centre for Genetic Engineering and Biotechnology (ICGEB), 34149 Trieste, Italy; Lorena@icgeb.org; 3Neuroscience Institute, National Research Council (CNR), 35131 Padua, Italy; 4Venetian Institute of Molecular Medicine (VIMM), 35131 Padua, Italy

**Keywords:** Alzheimer’s disease, hippocampus, mitochondria, membrane potential, Ca^2+^, excitotoxicity, Alisporivir

## Abstract

In Alzheimer’s disease (AD), the molecular mechanisms involved in the neurodegeneration are still incompletely defined, though this aspect is crucial for a better understanding of the malady and for devising effective therapies. Mitochondrial dysfunctions and altered Ca^2+^ signaling have long been implicated in AD, though it is debated whether these events occur early in the course of the pathology, or whether they develop at late stages of the disease and represent consequences of different alterations. Mitochondria are central to many aspects of cellular metabolism providing energy, lipids, reactive oxygen species, signaling molecules for cellular quality control, and actively shaping intracellular Ca^2+^ signaling, modulating the intensity and duration of the signal itself. Abnormalities in the ability of mitochondria to take up and subsequently release Ca^2+^ could lead to changes in the metabolism of the organelle, and of the cell as a whole, that eventually result in cell death. We sought to investigate the role of mitochondria and Ca^2+^ signaling in a model of Familial Alzheimer’s disease and found early alterations in mitochondria physiology under stressful condition, namely, reduced maximal respiration, decreased ability to sustain membrane potential, and a slower return to basal matrix Ca^2+^ levels after a mild excitotoxic stimulus. Treatment with an inhibitor of the permeability transition pore attenuated some of these mitochondrial disfunctions and may represent a promising tool to ameliorate mitochondria and cellular functioning in AD and prevent or slow down cell loss in the disease.

## 1. Introduction

Alzheimer’s disease is the most common form of neurodegenerative disease affecting millions of people worldwide (https://www.alzint.org/about/dementia-facts-figures/dementia-statistics/ (accessed on June 2021). Biological hallmarks of the disease are extracellular deposition of amyloid plaques, composed mainly by fibrils of amyloid-ß (Aß) peptide, intracellular neurofibrillary tangles (NFTs) consisting of aggregated, hyperphosphorylated tau protein, and extensive neurodegeneration [1]. Clinical symptoms are cognitive impairment and dementia characterized by progressive, profound disability culminating with death [2,3,4]; it is proposed that subtle alterations in the physiology of the brain and the so-called preclinical phase might precede the overt manifestation of clinical symptoms by 2–3 decades [5]. The dominant hypothesis for explaining AD alterations postulates that the aggregation of Aß peptides initiate the cascade of events leading to inflammation, the formation of tau-tangles, synaptic loss, and neurodegeneration [6,7]. However, this hypothesis has been challenged, mainly because numerous clinical trials aimed at decreasing Aß burden have failed to deliver significant benefits to the patients, and because there is not a strong correlation between phases of amyloid deposition and degree of cognitive decline, reviewed in [8], or regional brain hypometabolism and amyloid plaque burden [9]. The majority of AD cases are sporadic (SAD) with no defined etiology other than age, while about 1% of AD cases are genetic (familial AD, FAD) caused by mutations in three genes responsible for the synthesis and processing of the ß-amyloid precursor protein (APP). These are *APP*, *PSEN1*, coding for presenilin-1, and *PSEN2* coding for presenilin-2. The mutations are autosomal dominant and affected people present the common alterations of AD, often anticipating the onset of the disease by several years in comparison to the sporadic forms and showing a very rapid rate of progression. Although the familial cases are rare, their study has provided important clues that help the understanding of the pathology, both via the direct examination of affected individuals and also via the creation of animal models expressing the human mutated genes. We took advantage of a mouse model expressing both a mutated form of human PS2 and human APP (B6.152H transgenic line) [10,11] to investigate cellular alterations that might emerge early in the course of the disease and precede significant Aß deposition contributing to cellular impairment. Numerous mitochondria alterations have long been described in AD [12,13], ranging from a decrease in cytochrome c oxidase (COX) subunits described in the postmortem AD-brain [14], altered fission/fusion processes [15], and compromised mitophagy processes [16]. We evaluated mitochondria functions in primary hippocampal neurons derived from B6.152H newborn pups and observed a slight but consistent impairment in the capacity to sustain their primary functions during challenging circumstances i.e., in conditions of disablement of components of the respiratory chain, or when an extra workload imposes an increase in the respiratory capacity of the cells. In addition, treatment with low glutamate concentration induces rapid Ca^2+^ uptake in mitochondria, reflecting the cytosolic Ca^2+^ increase, which, compared to controls, in cells from the FAD mouse models, take longer to be cleared off from the matrix, and often is paired with sustained mitochondrial membrane depolarization. These subtle alterations could make B6.152H cells more vulnerable to toxic stimuli, or less capable to sustain intensive stimulations, and in the long run, this could compromise their fitness and survival.

## 2. Materials and Methods

### 2.1. Animal Handling and Care

The transgenic mouse line B6.152H was a kind gift from Dr. L. Ozmen (F. Hoffmann-La Roche Ltd., Basel, Switzerland). The line has the C57BL/6 strain background, and wild-type animals of this strain were used as controls. All procedures and experiments involving animals were conducted in accordance with the Italian and European Community Council Directive on Animal Care, and approved by the Italian Ministry of Health, protocol number: D2784.N.HEH.

### 2.2. Primary Neuronal Cultures

Primary neuronal cultures were isolated from the brain, the hippocampus region, of 0- to 1-day old newborn mice of either sex. Cells were seeded on poly-D-lysine (10 µg/mL), laminin (2 µg/mL)-coated coverslips at a density of 320,000 cells cm^−2^ in MEM based medium (Gibco 32360-026) supplemented with 20 mM Glucose, 1 mM Na-pyruvate; 1% N2 supplement (Thermo Fisher 17502048, Waltham, MA, USA), 9 µg/mL biotin, 0.5 mM L-glutamine, 0.5% B27 supplement (Thermo Fisher 17504044), 10% horse serum, 25 µg/mL penicillin, and 25 µg/mL streptomycin. The medium was replaced 48 h after seeding with the BME medium (Gibco 41010-026) containing 5 mM glucose and supplemented with 0.5 mM L-glutamine, 2% B27 supplement, and 1 mM Na-pyruvate. Experiments were performed after 14–17 days in vitro (DIV).

### 2.3. Adeno-Associated Virus (AAV) Production

Recombinant AAV vectors used in this study were prepared by the AAV Vector Unit at the International Center for Genetic Engineering and Biotechnology Trieste (http://www.icgeb.org/avucore-facility.html), as described previously [17,18]. Briefly, infectious, recombinant AAV vector particles were generated in HEK293T cell cultures grown in roller bottles, using a cross-packaging approach by which the vector genome is packaged into serotype-9 AAV capsid [19]. Viral stocks were obtained by PEG precipitation and subsequent cesium chloride (CsCl_2_) gradient centrifugation [20]. The titer of recombinant AAVs was determined by real-time PCR quantifying vector genomes (vg) against a standard curve of plasmid containing the specific vector genome [21], and the values obtained were in the range of 1 × 10^12^–1 × 10^13^ vg per milliliter.

### 2.4. Ca^2+^ Imaging

Cytosolic and mitochondrial Ca^2+^ measurements were performed using genetically encoded Ca^2+^ indicators (geci), jRGECO1 [22] (pAAV.Syn.NES-jRGECO1a.WPRE.SV40 Addgene plasmid #100854; http://n2t.net/addgene:100854; RRID:Addgene_100854) and 4mtD3cpv [23] (Addgene plasmid #36324; http://n2t.net/addgene:36324; RRID:Addgene_36324), respectively. A 4mtD3cpv probe was inserted into the pAAV.hSyn.eGFP.WPRE.bGH plasmid, at the sites Kpn1 and HindIII for virus production, after removing the eGFP coding sequence, (pAAV.hSyn.eGFP.WPRE.bGH Addgene plasmid #105539; http://n2t.net/addgene:105539; RRID: Addgene_105539) After 8–10 DIV, the cells were infected with approximately 1–2 × 10^9^ viral particles mL^−1^ and cultivated for further 6–4 days to ensure appropriate expression and folding of the probes. Ca^2+^ levels were monitored using an inverted fluorescence microscope Nikon Eclipse T*i* equipped with a Zyla-CMOS 4.2-P camera (Andor, Oxford Instruments, Abingdon, UK) and a 40× objective Nikon S Fluor Oil 1.30 DIC H/N2 WD 0.22). Excitation wavelengths were obtained with a 75 W Xenon Lamp (USHIO, UXLS50A) and a monochromator (Cairn Optoscan Monochromator, Cairn Research Ltd., Faversham, UK) controlled by NIS-E software (Nikon). Excitation wavelength for jRGECO1 was 535 nm, the beam passed first thorough an excitation filter 542/20 and a dichroic mirror FF-570-Di01, both from Semrock (Semrock Inc, IDEX Corporation, Lake Forest, IL, USA); the emitted light was collected through a Semrock BA 620/52 filter and the exposure time was 80 ms. 4mtD3cpv was excited at 425 nm, the beam passed first through a 438/20× filter (Semrock) and a dichroic mirror FF 458-Di02 (Semrock), and the emitted light passed first through a beam splitter (Optosplit T515LPXR-UF2, 301,699 Chroma) and was collected through two emission filters: FF01-479/40 for CFP and FF01-542/27 for YFP, (both filters from Semrock), and exposure time was 100 ms. The measurements were performed at 37 °C using a modified Krebs–Ringer buffer (mKRB) 140 mM NaCl, 2.8 mM KCl, 2 mM MgCl_2_, 2 mM CaCl_2_, 10 mM HEPES, 5 mM glucose, pH 7.4 at 37 °C, to mimic the brain extracellular fluid composition, the coverslips were mounted into an open-topped chamber (RC-41LP, Warner Instruments).

### 2.5. TMRM Experiment

The neuronal culture was incubated for 30 min at 37 °C with 10 nM TMRM, 1 μM cyclosporin H in a modified Krebs–Ringer buffer (mKRB) 140 mM NaCl, 2.8 mM KCl, 2 mM MgCl_2_, 2 mM CaCl_2_, 10 mM HEPES, 5 mM glucose, pH 7.4 at 37 °C. The coverslip was then transferred onto a thermostated chamber on the microscope stage and incubated for a further 10 min to verify signal stability. A combination of respiratory chain and ATP-synthase inhibitors was used to verify their impact on the mitochondrial membrane potential. TMRM fluorescence was recorded using 40× objective (SFluor 40× N.A. 1.3, Nikon, Minato, Tokyo, Japan) on an inverted microscope (Nikon Ti-E). Fluorescence illumination used a 50–75 WLamp (USHIO UXLS50A) and 550 nm excitation wavelengths was obtained using a monochromator (Optoscan CAIRN-Research, Faversham, UK) controlled by NIS- ELEMENTS AR (Nikon) software. The emitted fluorescence was collected using an FF-570- Di01Dichroic (Semrock, Lake Forest, IL, USA) and a 620/52 nm (Semrock) filter. Images were acquired every 60 s, with 50 ms exposure times by a Zyla-CMOS 4.2-P camera (Andor, Oxford Instruments, Belfast, Northern Ireland). Where indicated, rotenone (1 µM) or antimycin-A (1 µM) were added. At the end of each experiment, 4 µM FCCP was added to assess the correct distribution of the dye. Images were exported as TIFF, analyzed with ImageJ (National Institutes of Health).

### 2.6. Superoxide Measurements

The cells (14–17 div) were incubated for 10 min in mKrebsRB with 200 nM MitoSOX and 1 μM cyclosporin H (CsH) to ensure proper loading of the cells, were subsequently washed two times with plain mKrebsRB and maintained in mKrebsRB 1 μM (CsH). The coverslip was then transferred on the thermostated chamber and kept on the microscope stage for 20 min before beginning with the acquisition. Fluorescence was recorded using 40× objective (SFluor 40× N.A. 1.3, Nikon, Minato, Tokyo, Japan) on an inverted microscope (Nikon Ti-E). Fluorescence illumination used a 50–75 WLamp (USHIO UXLS50A) and the sample was alternately illuminated with 405 and 490 nm excitation wavelengths, obtained using a monochromator (Optoscan CAIRN-Research, Faversham, UK) controlled by NIS-ELEMENTS AR (Nikon) software. Excitation light passed through a QUAD turret emission filter FFO l-390/482/563/640. The emitted fluorescence was collected using a DÌ03-R405/488/561/635 dichroic mirror and an FFO l-446/523/600/677 filter (all from (Semrock, Lake Forest, IL, USA). Multiple images were acquired for every coverslip, with 100 ms exposure times by a Zyla-CMOS 4.2-P camera (Andor, Oxford Instruments, Belfast, Northern Ireland).

### 2.7. H_2_O_2_ Measurements

WT and B6.152H 7 D.I.V. cultures were transduced with approximately 6.5 × 10^13^ U/mL viral particles carrying the gene for hsyn-mito roGFP2orp1. At day 14–17 in vitro, H_2_O_2_ production was measured using an inverted fluorescence microscope Nikon Eclipse T*i* and a 40× objective Nikon S Fluor Oil 1.30 DIC H/N2 WD 0.22). Excitation wavelengths were obtained by a monochromator (Cairn Optoscan Monochromator, Cairn Research Ltd. UK). Cells were alternatively illuminated with 405 and 488 excitation wavelengths light passed through Semrock QUAD excitation filter FF01-390/482/563/640; emitted light first passed through the Semrock dichroic mirror Di03-R405/488/561/635, and 510 nm emitted light was collected through a Semrock emission filter FF01-446/523/600/677; the exposure time was 100 ms. The measurements were performed at 37 °C using a modified Krebs–Ringer buffer (mKRB) 140 mM NaCl, 2.8 mM KCl, 2 mM MgCl_2_, 2 mM CaCl_2_, 10 mM HEPES, 5 mM glucose, pH 7.4 at 37 °C, to mimic the brain extracellular fluid composition; acquisitions were performed every minute. Images were next analyzed using the ImageJ software. Regions of interest (roi) were manually drawn on neuronal cell bodies where mitochondria expressing roGFP2orp1 were clearly visible, and the ratio between light emitted for the 405 nm excitation and the 488 nm excitation was calculated.

### 2.8. Protein Extraction and Western Blotting

Proteins were extracted from 14–17 DIV neuronal culture using an extraction buffer: 150 mM NaCl, 50 mM Tris-Cl (pH 8.0), 1% Nonidet P-40, 0.5% sodium deoxycholate, 0.1% SDS with a complete mini, EDTA-free protease inhibitors cocktail (Roche Applied Science). The extracts were quantified using the BCA Protein assay kit (QuantumProtein Assay, EuroClone) as per manufacturer instruction. The protein suspension was dissolved in reducing Laemli sample buffer and heated for 5 min at 50 °C. Thirty micrograms of the protein extract was loaded per lane and separated by SDS-PAGE (12.5% polyacrylamide gel) and subsequently transferred onto nitrocellulose membrane (GE Healthcare #10600001) and probed using the following antibodies: MitoProfile total OXPHOS rodent WB antibody cocktail (Abcam ab110413) 0.6 µg/mL; mouse anti Hsp90 1:1000 (BD Bioscience #610418). Species-specific, HRP-conjugated secondary antibodies (BioRad) have been used. Immuno-bands were visualized using the chemiluminescence reagent Westar Sun (Cyanogen) on an UviTech Mini HD9 system (Eppendorf). Band intensities were analyzed using the image processing software ImageJ (NIH, USA), using the HSP90 signal for normalization.

### 2.9. Immunofluorescence

Fourteen- to seventeen-day in vitro hippocampal neurons were fixed with a medium containing 4% PFA and 20% sucrose dissolved in phosphate buffered saline (PBS), for 10-min, all subsequent procedures were conducted at room temperature. The coverslips were then washed 3 times, for 5 min each, with PBS and incubated with quenching solution (0.24% NH4Cl in PBS) for 20 min and washed again at the end of incubation. Cells were then treated with 0.1% TRITON-X 100 in PBS (permeabilization solution) for 3-min, and subsequently incubated with 10% Goat serum, 2% BSA, 0.2% gelatin, in PBS for 30 min to block unspecific reaction sites and then incubated for 1 h with mouse anti-neurofilament 200 (1:100, Merck-Sigma N5389); rabbit anti-glial fibrillary acidic protein (1:100 Dako, Z0334). After rinsing, the cells were incubated for 45 min with Alexa Fluor secondary antibodies Fluor Plus 555 goat anti mouse (1:500 Thermo Fisher Scientific A32727), and Alexa Fluor Plus 488 goat anti rabbit (1:500 Thermo Fisher Scientific A32731. To mark the nuclei, the cells were incubated for 10 min with 5 µg/mL Hoechst 33,342 (Invitrogen H1399) and mounted on glass coverslips using Mowiol 4–88 (Millipore 475904-M). Images were collected with the Nikon Eclipse 80i ViCo fluorescence microscope system, equipped with a Qimaging Fast 1394 camera and subsequently elaborated with the image processing software ImageJ (NIH, USA).

## 3. Results

### 3.1. FAD Hippocampal Neurons Have Impaired Respiratory Capacity

Mitochondria dysfunction has long been implicated in the cellular alterations that characterize AD. To further address this issue, and with the aim of carefully probing mitochondria metabolism in AD, we used primary cortical neurons isolated from the hippocampi of newborn WT and B6.152H mice. This last genotype represents a model for Familial Alzheimer’s disease as it expresses the mutated human presenilin 2 (PS2-N141I) protein, together with the human mutated (Swedish variant K670N, M671L)Amyloid Precursor Protein (APP) [10,11]. Neuronal cells derived from the hippocampi of newborn pups, p0–p1, were grown for 14–17 days in vitro (div) to allow the expression of membrane channels and receptors, in a medium with a physiological glucose level (5 mM), to avoid potential toxic effects of the high glucose media commonly used [24], and to maintain a balance between aerobic and anaerobic metabolism in the cultured neurons. For all the following experiments with neurons, we seeded the same number of cells for both WT and B6.152H, checked the cells at regular intervals to control for possible contamination and to replenish consumed medium, and we did not observe differences in the growth rate or cell death between the two genotypes. The oxygen consumption rate (OCR) was measured in the cultures by means of the Seahorse XF Analyzer as described in [25]. Basal OCR and ATP synthesis-driven OCR (oligomycin-sensitive respiration) were comparable (not statistically different) in both WT and B6.152H neuronal cultures, while maximal OCR induced by the uncoupling of oxidative phosphorylation (achieved with the addition of the protonophore carbonyl cyanide 4-trifluoromethoxy-phenylhydrazone, FCCP) (relevant substances used in this work are listed in Table 1, Section 2) was significantly reduced in B6.152H neurons compared to WT (Figure 1). Since sometimes the addition of oligomycin inhibits maximal respiration in the cells [26], the addition of FCCP was performed in separate wells from oligomycin, or, when added after oligomycin, only the wells at least doubling OCR with respect to basal OCR were considered.

Simultaneously, in the same plates, we also quantified the extracellular acidification rate, ECAR, which is an indirect measurement of the glycolytic flux of the cells. The main contributor to ECAR is the lactic acid produced by glycolysis, but other cellular metabolic pathways can change the extracellular pH mostly by producing CO_2_, namely the TCA cycle. In our experimental conditions, B6.152H cultures showed a lower basal ECAR, compared to wild-type cells, which could be an indication of a glycolytic defect as already suggested for B6.152H cortical neurons [25]. Maximal ECAR is measured after the addition of oligomycin (2 μg/mL), which inhibits F_1_F_0_ ATP synthase activity, blocking mitochondrial ATP synthesis and shifting cellular energy production to glycolysis. B6.152H hippocampal neurons had significantly lower maximal ECAR than WT neurons, suggesting a possible diminished capacity for these cells to meet energy challenges.

### 3.2. FAD Hippocampal Neurons Are Less Able to Maintain a Stable Membrane Potential When Challenged with Inhibitors of the Respiratory Chain

On the day of experiment, the cells were incubated for 30 min at 37 °C with 10 nM TMRM supplemented with 1 μM Cyclosporin H (CSH) to ensure stable loading of the dye (Figure 2A). The coverslip was then transferred onto a thermostated chamber on the microscope stage and incubated for another 10 min to equilibrate the temperature and prevent focal drift due to thermal imbalance. As shown in (Figure 2B), the cultures are enriched in neurons but contain a mixed population of neurons and glial cells. For TMRM measurements, and for all subsequent imaging experiments, only neurons were examined, placing the region of interest (roi) on the soma of cells previously identified as neurons by their morphology in bright field images, round-shaped somata, and distinct processes [27], or by the specific targeting of genetically encoded sensors (see below).

The basal TMRM fluorescence levels between WT and B6.152H neurons were comparable (Figure 3 inset), indicating a similar loading of the dye, and excluding the possibility of conspicuous differences in plasma membrane or mitochondria membrane potential between the two genotypes, and allowing us to normalize the fluorescence data to the average of the initial 5 min acquisitions. Cultured neurons were then challenged with drugs known to inhibit the mitochondrial respiratory chain specifically at the level of complex I (rotenone) or complex III (antimycin), with the aim of identifying hidden defects in the capacity of mitochondrial ATP-synthase to maintain the inner membrane potential via hydrolysis of ATP molecules within the time frame of the experiment. As shown in Figure 3A, upon addition of rotenone mitochondria of B6.152H, cells lost at least part of their membrane potential much faster than WT mitochondria. This result could be attributed either to a defect in ATP-synthase activity, or/and to a reduced availability of ATP. Similar results were observed inhibiting complex III with antimycin. On the contrary, the cells responded in a similar way to the blockade of the catalytic activity of ATP-synthase with oligomycin (Figure 3B); we observed a slight initial depolarization and a small hyperpolarization in B6.152H cells, but overall, the cells maintained a stable mitochondrial membrane potential for the whole duration of the experiment, indicating that the respiratory complexes of the two genotypes are equally able to sustain analogous membrane potential within the experimental timeframe. The slight depolarization observed after oligomycin addition might be due to a modest inhibition of the plasma membrane Na^+^/K^+^ATPase by the drug [28,29]. Figure 3C compares the decrease of TMRM fluorescence at 20 min after the addition of either respiratory chain inhibitor, and prior to the addition of the uncoupler (FCCP). The protonophore FCCP is added to completely collapse the mitochondrial membrane potential (providing a minimal value for the normalization of the data), and it also allows for controlling the TMRM-specific signal.

The ATP-synthase complex has been shown to be reduced in several post-mortem brain samples of Alzheimer’s disease patients [30], and defects in the enzyme have also been described in an AD mouse model [31]; a decreased expression of several nuclear encoded ATP-synthase genes have been found in brain areas early affected by AD [32]. To investigate possible ATP-synthase defects in our experimental model, we measured the total ATP content of the culture using a commercial kit, as per manufacturer instructions (Abcam Luminescent ATP detection assay kit ab113849) and did not find an appreciable difference between WT and B6.152H cells (results not shown). It needs stressing, however, that not only we have evaluated total ATP contributed by all the cell types present in the culture, and not just neurons [32], but also, and most relevant, that part of the cellular ATP is produced by glycolysis. To more directly evaluate whether the functioning of the mitochondrial ATP-synthase is impaired in the AD transgenic model, we measured ATP-synthase reverse activity in mitochondria derived from the brain of 3- and 6-month old WT and B6.152H mice, a protocol adapted from [33]. No appreciable differences were observed in this case between the two genotypes (Figure 4), but also in this case, the mitochondria are derived from whole brain tissue (mainly from cortical and hippocampal areas).

### 3.3. Mitochondrial Reactive Oxygen Species Production

A large body of scientific literature points to the effect of reactive oxygen species (ROS) in the progression and possibly the onset of neurodegenerative disorders, and mitochondria dysfunctions have been examined as possible early events in the pathogenesis [34,35], with mitochondria being important ROS producers within the cells. It has been shown that the malfunctioning of the mitochondrial respiratory chain, impairing the flux of electrons (e^−^) from complex I to final O_2_ reduction to H_2_O at the level of complex IV, often results in the production of reactive oxygen species due to the leak of e^−^ from reduced cofactors. The anion radical superoxide (O_2_^−•^) is produced by direct reaction of a single e^−^ with molecular O_2_, and the excess of matrix superoxide can initiate oxidative damage. The dye hydroethidine and its mitochondrial derivative MitoSOX™ are commonly used to measure O_2_^−•^ in living cells. MitoSOX™ accumulates within the mitochondrial matrix driven by the inner membrane potential. Given that in resting conditions the membrane potential is comparable between WT and B6.152H neurons, the MitoSOX™ signal appears adequate to measure basal O_2_^−•^ production in mitochondria of WT and B6.152H neuronal cultures. No significant differences were observed however (Figure 5). Often, inhibitors of the respiratory chain are added to increase ROS production and to better visualize it in imaging experiments We could not exploit MitoSOX™ to assess O_2_^−•^ in conjunction with rotenone and antimycin inhibition of the respiratory chain, since such treatments cause the depolarization of mitochondria, which in the end lead to exit of the dye from the matrix and to its redistribution in the cytoplasm.

Mitochondrial O_2_^−•^ is rapidly detoxified to the less reactive hydrogen peroxide (H_2_O_2_) by matrix Manganese Superoxide dismutase (Mn-SOD). To measure H_2_O_2_ production, we used a genetically encoded, mitochondrially targeted probe named mito-ro-GFP2-orp1 (Addgene #65001) [36] modified by adding the human synapsin 1 promoter to achieve specific expression in neurons; nine div neurons were infected with adeno-associated virus (AAV) carrying the mitochondria targeted ROS probe. In our experimental settings, we could not detect any increase of H_2_O_2_ upon inhibition of the respiratory chain with rotenone or antimycin (results not shown); in addition, the basal ratio fluorescence of the probe was comparable between WT and B6.152H neurons (Figure 6), as seen with MitoSOX™ probe, meaning that either in our cultures, ROS production is quite low and efficiently counteracted by the intracellular antioxidative mechanisms, or that the sensitivity of the system is too low to detect subtle differences.

### 3.4. FAD Hippocampal Neurons Are More Sensitive to Glutamate Excitotoxicity

#### 3.4.1. Effect of Glutamate on Cytosolic Ca^2+^ Transients

Glutamate is considered the principal excitatory neurotransmitter in the brain [37], important for the fundamental memory and learning processes [38,39]. Its concentration has to be maintained within tight boundaries for proper signal transmission, and to avoid toxic effects for the cells. Moreover, excitotoxicity has been proposed as a concurrent mechanism for the development of neurodegenerative disorders together with soluble amyloid beta (Aβ) oligomers [40] and tau protein hyperphosphorylation [41]. In order to induce a mild excitotoxic stress to the neurons and to evaluate their responses, we treated 14–17 div WT and B6.152H hippocampal neurons with a low concentration of glutamate (2 μM), which is one order of magnitude above the basal extracellular neurotransmitter concentration [42]. Glutamate is known to induce an increase in intracellular [Ca^2+^] due to a complex series of events: (i) Activation of AMPA receptors with depolarization and Ca^2+^ influx through voltage gated channels; (ii) opening of ionotropic NMDA receptors; and (iii) activation of Group I mGlu receptors. To evaluate the variations of intracellular Ca^2+^ content in cultured neurons after the toxic stimulus, WT and B6.152H neurons, 9 div, were infected with AA-virus expressing the cytosol targeted Ca^2+^ probe jRGECO [22] under the human synapsin 1 promoter. The cells were then examined 5 days post-infection to allow complete expression and folding of the probe. Basal fluorescence was comparable between WT and B6.152H neurons, indicating a similar expression of the probe between the two genotypes; as shown in Figure 7, we could sometimes observe spontaneous, synchronous Ca^2+^ oscillations in the resting cells of both genotypes. jRGECO is not a ratiometric probe, and accordingly, the intensity of the signal does not allow one to directly correlate it to [Ca^2+^]. Although a ratiometric dye would be easier to calibrate in terms of [Ca^2+^], we chose jRGECO for these experiments for its fluorescence intensity and suitability to detect the Ca^2+^ changes due to neuronal action potentials [22]. Information on the amplitude of the [Ca^2+^] changes with a non ratiometric probe can, however, be obtained by the measurements of the ΔF/F_0_ values. Being non ratiometric, in addition, means the probe is potentially affected by the pitfalls or artefacts due to movements of the focal plane or of the cells during the acquisitions, so extra care was taken to ensure stability of the focal plane, and movements of the cells were controlled for during offline analysis. The cells were challenged with the addition of 2 µM glutamate to mimic a sustained excitatory signal that elicited a large cytosolic Ca^2+^ increase. The rate of fluorescence increases, and the peak amplitude reached were comparable between the two genotypes, with no significant difference, indicating analogous endowment of cellular receptors and channels. After the rapid Ca^2+^ rise, its concentration progressively decreased to a lower plateau level; we then measured the jrGECO ΔF/F_0_ ratio 20 min post glutamate addition to investigate whether the cells were equally able to extrude excess Ca^2+^ from the cytosol and return toward the basal level. In the time frame of the experiment, we observed that both cell types reached a comparable cytosolic [Ca^2+^]. As discussed above, this probe does not allow the quantification of the absolute values of cytosolic [Ca^2+^] in the neurons, but the calculation of the ΔF/F_0_ permits one to compare the variation in [Ca^2+^] produced by the stimulus in the two cell types. Potassium chloride (KCl) 30 mM was added at minute 25, towards the end of acquisition, to further depolarize the plasma membrane, promote the opening of Voltage-Operated Channels (VOCs), and evaluate the residual capacity of the cells to accumulate Ca^2+^.

#### 3.4.2. Effect of Glutamate on Mitochondrial Ca^2+^ Transients

Mitochondria actively participate in the control of cellular Ca^2+^ content, helping in finely shaping Ca^2+^ signals [43,44]; in addition, the activity of the respiratory chain is modulated by matrix Ca^2+^ due to its effects on key mitochondrial dehydrogenases. In the next series of experiments, we investigated the [Ca^2+^] changes within the mitochondrial matrix brought about by the glutamate challenge. For these experiments, WT and B6.152H neurons were infected with AAV expressing the mitochondrial Ca^2+^ probe 4mtD3cpv [23] under the control of the human synapsin 1 promoter. To ensure complete expression and folding of the probe, the cells were infected at least 5–7 days prior to the experiment. In the case of 4mtD3cpv, the ratio between the fluorescence emitted at 520 and 480 nm is directly proportional to the [Ca^2+^], independently of the amount of probe expression level [23,45]. The cells of the two genotypes have comparable mitochondrial resting Ca^2+^ levels (Figure 8 panel C). As before, the cells were then challenged with 2 μM glutamate in the presence of 2 mM external Ca^2+^. As expected, a rapid and large rise of matrix Ca^2+^ was observed upon glutamate addition. The first part of the response was generally homogeneous, with the majority of cells of both genotypes reaching a very similar Ca^2+^ peak with the same rate, suggesting that the functioning of the mitochondrial calcium uniporter (MCU) is comparable between the genotypes. The fluorescence of 4mtD3cpv was then monitored for about 25 min after glutamate addition and it was observed that after the first fast mitochondrial Ca^2+^ uptake, the behavior of the cells was quite variable. In some cells, the Ca^2+^ concentration steadily decreased toward basal, while in others, the Ca^2+^ inside the mitochondria begun to oscillate, giving rise to waves of Ca^2+^ that were very rarely observed in resting conditions. These oscillations were more frequently observed in mitochondria of B6.152H cells, 27.3% of the total, compared to 20% of all observed WT cells, and were clearly not attributable to changes in the focal plane, or movements of the cells, since 4mtD3cpv is a ratiometric probe that allows one to correct for these artifacts. Finally, the cells were challenged with the addition of 30 mM KCl, to produce a sustained plasma-membrane depolarization and prolonged opening of voltage activated Ca^2+^ channels. Mitochondria responded to KCl with a large Ca^2+^ peak that was comparable to that obtained with the addition of glutamate. We calculated the ratio between YFP and CFP fluorescence for 4mtd3cpv after 20 min from glutamate addition in order to measure the residual Ca^2+^ concentration in mitochondria after the stimulation. An appreciable difference between the Ca^2+^ level of WT and B6.152H cells was observed, with transgenic mitochondria showing a significantly higher level of Ca^2+^ (Figure 8 panel F). The inability of mitochondria to efficiently clear matrix [Ca^2+^] could be detrimental for the organelles, since a prolonged increase in matrix Ca^2+^ concentration could keep some of the Ca^2+^-dependent mitochondrial dehydrogenases active, and could also sensitize/prime the mitochondrial permeability transition pore (PTP) and promote its opening in the presence of a second cue (e.g., oxidation, changes in matrix pH, presence of toxic oligomeric species) [46,47].

### 3.5. Glutamate Excitotoxicity Impairs FAD Hippocampal Neurons Membrane Potential, Effect of Alisporivir

To evaluate the ability of the cells to maintain the membrane potential in the presence of a large and sustained Ca^2+^ influx, the neuronal cultures were challenged with 2 µM glutamate, in the presence of physiologic extracellular Ca^2+^. Figure 9A shows representative traces of TMRM fluorescence measured in WT and B6.152H hippocampal neurons treated with glutamate; fluorescence measurements (arbitrary units) were normalized to the difference between the average of initial fluorescence, minus the average of fluorescence signal measured after the addition of the protonophore FCCP. We observed that WT neurons can better sustain the excitotoxic challenge posed by glutamate, while double transgenic neurons seem to be more sensitive to the treatment, and a large proportion of cells lost their membrane potential faster than WT ones, as a consequence of the treatment.

The loss of membrane potential caused by the addition of glutamate and the increase of matrix [Ca^2+^] could be due to the incapacity of the respiratory chain to work at a sufficient rate when challenged with a sustained ion entry. However, it could also be caused by the opening of the permeability transition pore. We checked whether the PTP was involved in the depolarization caused by the glutamate excitotoxicity by treating B6.152H neurons with a low concentration (1.5 μM) of the inhibitor Alisporivir (formerly known as Debio 025), which inhibits PTP opening via cyclophilin D binding, but, differently from its parent compound, cyclosporin A, does not bind to calcineurin [48,49]. Cells were treated with 1.5 μM Alisporivir for 5 min prior the additions of glutamate. We controlled for the fact that Alisporivir per se does not change the basal membrane potential, and as seen in Figure 9, panel B, we did not observe any difference in basal fluorescence between WT, B6.152H neurons, and B6.152H neurons treated with the drug. On the other hand, Alisporivir treatment delayed, and often significantly reduced, the drop of membrane potential in B6.152H neurons challenged with glutamate (Figure 9 panel C). Alisporivir also had a mild effect on WT neurons, slightly increasing basal TMRM fluorescence, compared to untreated cells.

## 4. Discussion

In this study, we reported subtle alterations in mitochondrial functionality in cultured hippocampal neurons derived from a mouse model of AD, which could make neurons more susceptible to challenging conditions, or contribute to the accumulation of stress-induced modification that can make them more vulnerable. We observed a decrease in maximal respiratory rate (OCR) in intact hippocampal neurons isolated from newborn B6.152H pups carrying the human proteins APP and PS2 bearing mutations responsible for familial AD. Basal and ATP-linked OCR were comparable between WT and B6.152H cells, suggesting that in normal conditions, these cells carry out regular metabolic functions. The defect instead became apparent when the cells were challenged with a stimulus that increases the rate of the respiratory chain, which could mimic the necessity to increase the rate of ATP production [50,51]. Parallel to this, a decrease in basal and maximal glycolytic flux was also measured, which could exacerbate the consequences for the cell of coping with a stimulus increasing the cellular workload and the need for additional ATP production [25].

Defects in mitochondrial functionality or molecular endowment have been consistently reported for AD patients and also AD animal models [52,53,54], while mitochondrial F1Fo ATP synthase and single components of the respiratory chain have been found defective in both AD patients and AD models [30,31,32,55]. We did not observe a reduction in single respiratory components of the ETC or ATP-synthase (Appendix A): Protein extract obtained from four independent WT and B6.152H neuronal cultures were probed with antibodies against individual components of the ETC and the F_1_F_O_ ATP-synthase and normalized against the HSP90 protein, employed here as a housekeeper and as loading control. Nevertheless, the defect could lay in the organization of mitochondrial supra-molecular complexes, which are thought to be fundamental in the efficient e^−^ transport and H^+^ translocation, and for the maintenance of the mitochondria cristae structure [56,57,58]. In addition, if the defect lays specifically in one cell type it would be difficult to spot it with this approach since the cultures used are enriched in neuronal cells but also bear other cell types.

The potential small defects at the level of the electron transport chain (ETC) or the ATP-synthase were confirmed by monitoring the mitochondrial membrane potential of the neurons challenged with different, specific inhibitors. A faster dissipation of mitochondrial membrane potential was indeed observed in B6.152H neurons compared to controls when the electron flux was blocked either at the level of complex I or complex III. This effect would be indicative of a potential defect in the working of the ATP-synthase functioning in “reverse mode”, hydrolyzing ATP to maintain the membrane potential [59], or to an insufficient supply of ATP to the enzyme. Measurements of ATP-synthase activity or total ATP content did not highlight any difference between WT and B6.152H genotypes, but this could be due to the limits of the experimental approach.

The latent defect observed in this experimental model could become detrimental in conditions of increased energy demand i.e., to restore ion gradients across the plasma membrane during periods of intense signaling, or to cope with toxic stimuli. Excess glutamate is known to cause excitotoxic damage to the brain, which can end up in terminal cellular disfunction and cell death, and has been implicated in the neurodegeneration characteristic of AD [60,61,62]. In our experimental model, a low micromolar glutamate addition produced a rapid and transient Ca^2+^ increase in the cytosol and in the mitochondrial matrix of both WT and B6.152H neurons, followed by a recovery towards resting values. B6.152H mitochondria took longer to extrude excessive matrix Ca^2+^. Excessive [Ca^2+^] could be detrimental for mitochondria [63], ending up reducing ATP synthesis [64,65] and inhibiting complex I activity [66]. In addition, a large and persistent load of matrix Ca^2+^ could sensitize the permeability transition pore (PTP) favoring its open state [67].

A physiological increase in cytosolic Ca^2+^ produces a depolarization of the mitochondrial membrane potential, likely caused by Ca^2+^ cycling across the mitochondrial inner membrane [68,69], but a toxic extracellular glutamate increase provokes the decrease in mitochondrial membrane potential and also a perturbation of Ca^2+^ homeostasis [70]. Moreover, a large and persistent decrease of mitochondrial membrane potential would make mitochondria unable to provide sufficient ATP for the cellular needs, making it more vulnerable to internal and external stressors and potentially setting in motion the series of event leading to cell death [71]. Upon the addition of glutamate, we observed different responses from the cultured neurons with some cells keeping their basal membrane potential unchanged and others experiencing different degrees of depolarization. On average, the B6:152H cells experienced a larger decrease in membrane potential than WT neurons, 20 min after glutamate addition. Paradoxically, the average [Ca^2+^] within mitochondria is higher in B6.152H cells than in controls, despite no significant average difference in cytosolic [Ca^2+^] and a larger depolarization of the organelles in the transgenic cells. It needs to be stressed that these results are the average of several tens of cells, while it has been impossible to monitor Ca^2+^ levels (either cytoplasmic or mitochondrial) and mitochondrial membrane potential in the same cells. Accordingly, in cells with dramatic drops in mitochondrial membrane potential, the matrix [Ca^2+^] will not decrease to the basal level, but it will simply equilibrate with the cytoplasmic [Ca^2+^] i.e., it will be close to that in cells with normal mitochondrial membrane potential. On the contrary, in cells with only partial depolarization, the efficacy of Ca^2+^ efflux from the mitochondria via the Na^+^/Ca^2+^ exchanger (NCLX, exchanging 3 Na^+^ for 1 Ca^2+^) would be reduced [72], and the matrix Ca^2+^ would be higher compared to controls The average [Ca^2+^] in the mitochondrial matrix will thus depend on the percentage of B6.152H cells with the membrane potential identical, lower, or even completely depolarized compared to the controls. Most relevant, however, preventive incubation of the cells with a micromolar concentration of an inhibitor of the PTP delayed, and in some cases completely avoided, the decrease in the mitochondrial membrane potential suggesting that the PTP might be involved, and inferring that in the AD model, it could be more sensitive to stimuli promoting its opening. We judged the potentially protective effect of Alisporivir very interesting as it could be developed into a therapeutic tool for AD, and in vivo experiments on the FAD mouse model are presently being started by our group.

## Figures and Tables

**Figure 1 cells-10-02046-f001:**
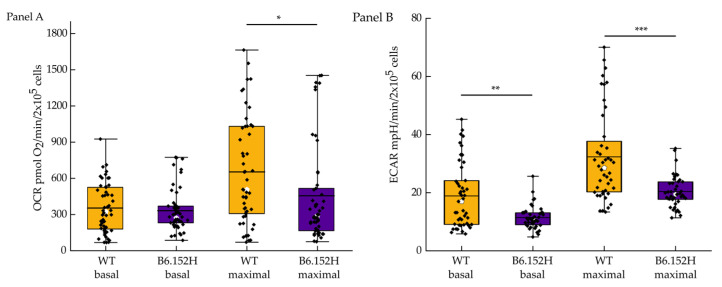
Quantification of intact-cell respiratory rate, OCR, and of the extracellular acidification rate ECAR. Panel (**A**), Basal OCR, (expressed as pmols of O_2_ consumed per minute per 2 × 10^5^ cells/well) combination of ATP synthesis-driven respiration plus the respiration caused by the leak of protons from the membrane (WT *n* = 50 wells, mean ranks = 57.88; B6.152H *n* = 50 wells mean ranks = 43.12; Mann–Whitney probability test Asymp. prob = 0.8); at the 0.05 level, the two distributions are not significantly different. Maximal OCR (uncoupled respiration), (WT *n* = 50 wells, mean ranks = 51.2; B6.152H *n* = 50 wells, mean ranks = 49.8; Mann Whitney probability test Asymp. prob = 0.011, at the 0.05 level the two distributions are significantly different. (WT, yellow; B6.152H, indigo). Panel (**B**), ECAR (expressed as mpH per minute per 2 × 10^5^ cells/well) quantification: Basal values WT *n* = 47 wells, median: 16.97. B6.152H *n* = 42 wells, median: 10.82. At the 0.05 level. the two populations are significantly different, Mann–Whitney *U* test, asymp prob = 0.005. Maximal ECAR, after 2 μg/mL oligomycin addition, WT median: 28.5, B6.152H median: 23.77. At the 0.05 level, the two populations are significantly different, Mann–Whitney *U* test, asymp. prob. = 4.6 × 10^−5^. Box representing 25–75% of data, whiskers are 1–99% of data, white square represents the mean value; transverse line represents the median value. Cells were from at least 4 different culture preparations. * *p* ≤ 0.05, ** *p* ≤ 0.01, *** *p* ≤ 0.001.

**Figure 2 cells-10-02046-f002:**
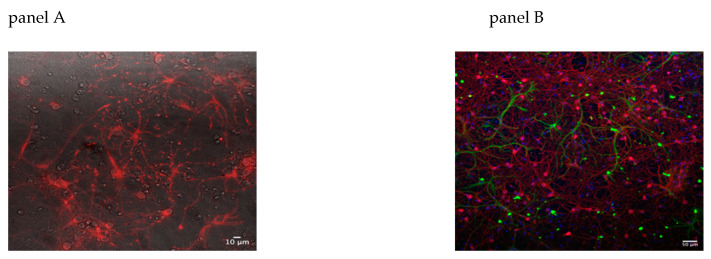
TMRM staining and immunofluorescence of mixed hippocampal mouse cultures. Panel (**A**), 15 div WT neuronal culture loaded with potentiometric dye TMRM 10 nM, imaged with the Nikon Eclipse T*i* inverted fluorescence microscope, 40× magnification; fluorescence image superimposed on the bright field image used to select neurons. Panel (**B**), representative image of 15 div neuronal culture; paraformaldehyde fixed cells were stained with: 1:100 anti-neurofilament 200 (NF 200, Merck-Sigma # N5389) labeling neurons, red; and with 1:100 anti glial fibrillary acidic protein (GFAP, Dako # 20334) labeling glial cells, green, and counterstained with Hoechst 100 μg/mL, blue, to mark all nuclei. Images were collected with a Leica DMI 6000 inverted microscope, equipped with ORCA FLASH 4.0 camera (Hamamatsu), magnification 20×. Analysis of images and composition of panels performed with the software ImageJ.

**Figure 3 cells-10-02046-f003:**
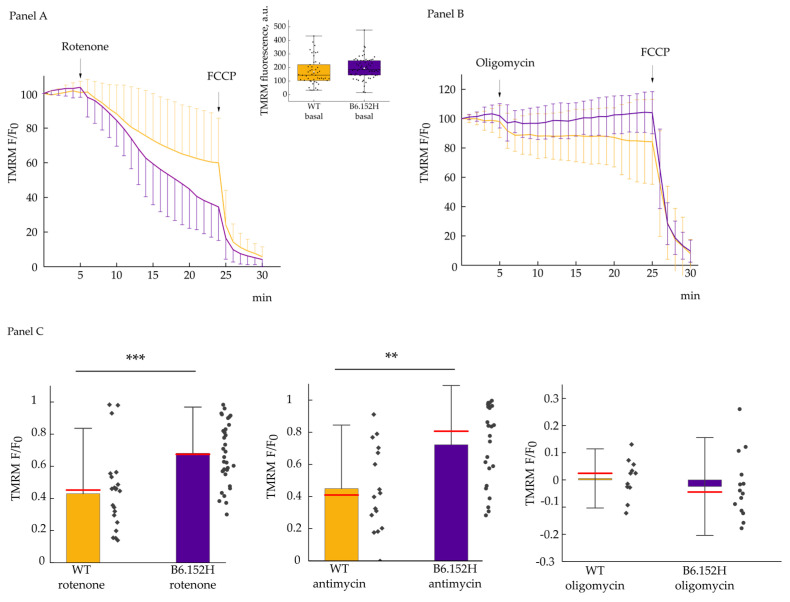
Effect of the inhibition of the ETC or of the F_1_F_0_ATP-synthase on the mitochondrial membrane potential. Panel (**A**) shows traces of TMRM fluorescence for WT, yellow, and B6.152H, indigo, neurons. Where indicated, the cells were challenged with 2 μM rotenone and 4 μM FCCP. Values are normalized over the mean fluorescence of the first five acquisitions; fluorescence intensity was acquired every min. Inset: Absolute TMRM initial values for WT, (yellow) *n* = 51 cells, and B6.152H (indigo) neurons *n* = 67 cells; test *t p* = 0.11. Box representing 25–75% of data, whiskers: 1–99% of data, white square: Mean; transverse line: Median. Panel (**B**) shows mean traces of TMRM fluorescence of WT and B6.152H neurons challenged with 1 μg/mL oligomycin and 4 μM FCCP. Panel (**C**) shows the quantification of the effects produced on the mitochondrial membrane potential by the above-mentioned inhibitors expressed as the normalized difference between initial fluorescence F0, minus the fluorescence measured at 25 min,. Rotenone experiment WT *n* = 22, B6.152H *n* = 34 *p* = 1.5 × 10^−4^ test *t* (normal distribution); antimycin experiments: WT *n* = 16, B6.152H *n* = 24 *p* = 0.003 Mann–Whitney test. Oligomycin experiments: WT *n* = 11, B6.152H *n* = 14 *p* = 0.48, test *t* (normal distribution). Cells were derived from at least three independent preparations, both for WT and the B6.152H genotype. Columns represent mean value plus/minus standard deviation; red line represents the median of the values. ** *p* ≤ 0.01, *** *p* ≤ 0.001.

**Figure 4 cells-10-02046-f004:**
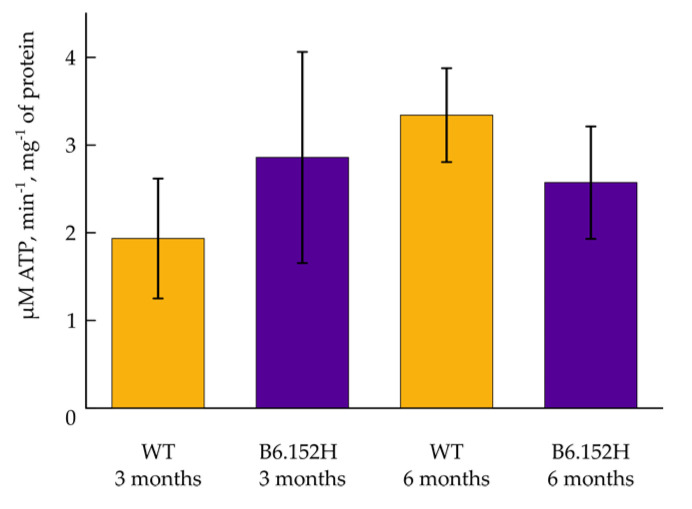
Measurements of the F1F0 ATP-synthase activity in isolated brain mitochondria. Quantification of the amount of ATP hydrolyzed per minute per mg of mitochondrial protein extracted from the forebrain of WT and B6.152H mice of 3 and 6 months of age. Mean of multiple probing from at least three independent mitochondria preparation for each genotype and for each age. For *p* ≤ 0.05, the differences among phenotypes were not statistically significant. Columns represent mean value plus/minus SEM.

**Figure 5 cells-10-02046-f005:**
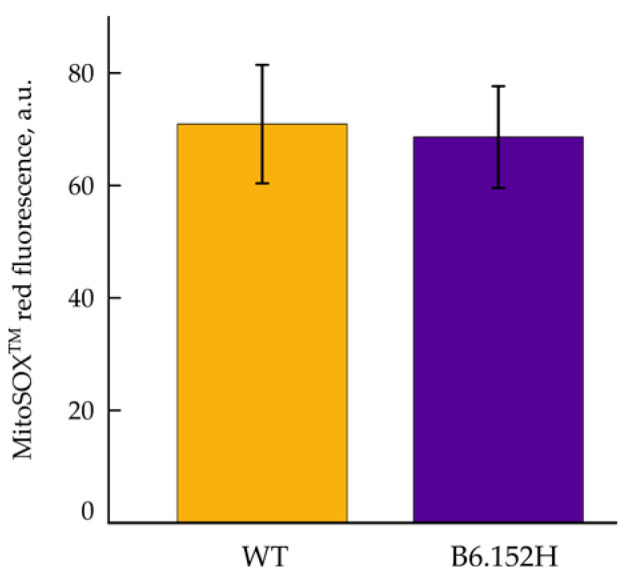
Measurements of superoxide production in primary hippocampal neurons. Basal MitoSOX fluorescence. WT, yellow, and B5.152H, purple, hippocampal neurons loaded with 200 nM MitoSOX™. The result is expressed as the average fluorescence for 8 WT and 6 B6.152H coverslips, respectively, with multiple neurons, derived from at least three independent preparations. For *p* ≤ 0.05, the differences between the two genotypes were not statistically significant. Columns represent mean value plus/minus SD.

**Figure 6 cells-10-02046-f006:**
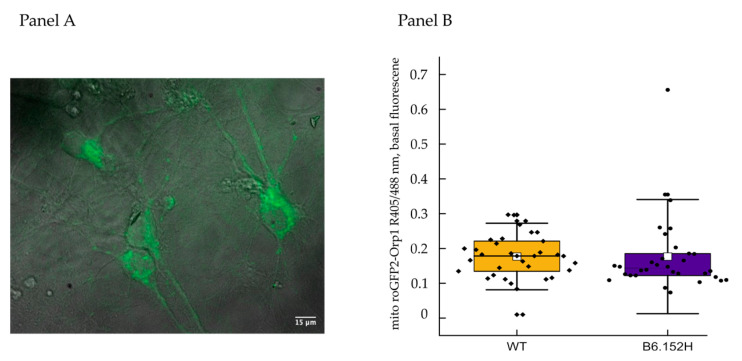
Measurements of H_2_O_2_ production in primary hippocampal neuronal cultures. Panel (**A**): Representative image of hippocampal neurons expressing the mitochondrial targeted probe for H_2_O_2_, mito roGFP2-orp1. Fluorescence image recorded illuminating the cells with a 488 nm excitation wavelength and superimposed to the corresponding bright field image. Panel (**B**), quantification of the basal fluorescence of the probe expressed as the ratio between 405 and 488 nm illuminations. Symbols represent individual cells obtained from at least three independent preparations for each genotype, WT, *n* = 34 cells, median = 0.18; B6.152H *n* = 32 cells, median = 0.14. Box representing 25–75% of data, whiskers: 1–99% of data, white square: Mean; transverse line: Median. For *p* ≤ 0.05, the difference between the two genotypes is not statistically significant, Mann–Whitney probability test, asymp. prob. = 0.17.

**Figure 7 cells-10-02046-f007:**
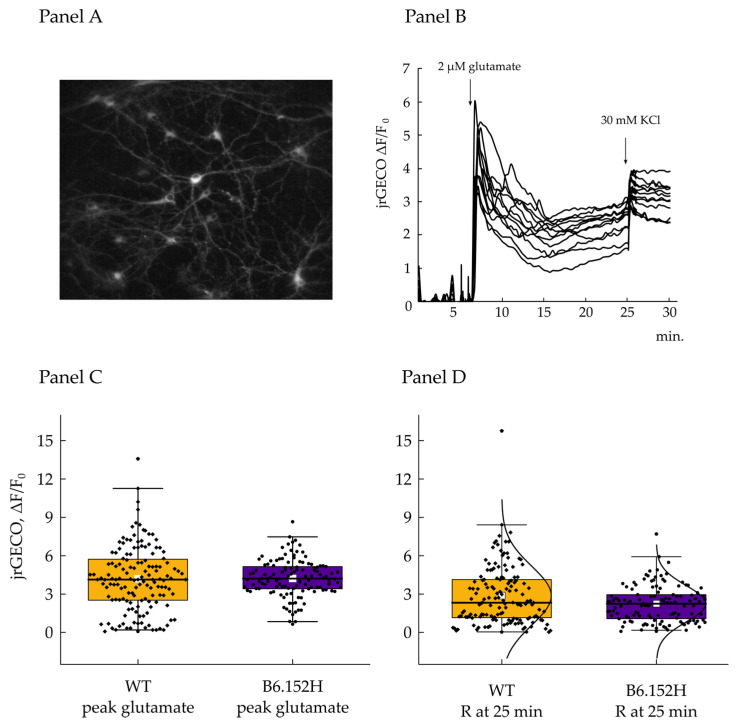
Cytosolic Ca^2+^ transient in primary hippocampal neurons induced by glutamate. Panel (**A**), representative image of 14 div hippocampal neurons expressing cytoplasm-targeted jrGECO. Panel (**B**), representative traces of jrGECO fluorescence expressed as the normalized values of the difference between time point F minus the average of the initial 5 min of acquisition (F_0_); where indicated the cells were challenged with 2 μM glutamate and 30 mM KCl. Panel (**C**), quantification of the jrGECO fluorescence measured at the maximum peak reached after glutamate addition and expressed as normalized difference with initial fluorescence F_0_. WT *n* = 144 cells, median value = 4.13; B6.152H *n* = 121 cells, median value = 4.2. Two sample t-test prob. = 0.9; at the 0.05 level, the difference between the means is not significantly different from 0. Panel (**D**), jrGECO fluorescence measured 20 min after glutamate addition and expressed as before. WT *n* = 144, median value = 2.33; B6.152H *n* = 121 cells, median = 2.24; Mann–Whitney test Asymp prob = 0.1. At the 0.05 level, the two distributions are not significantly different. Box plot: Box height, 25–75%; whiskers, 1–99%; white square: Mean; transverse line: Median; curve: Normal distribution.

**Figure 8 cells-10-02046-f008:**
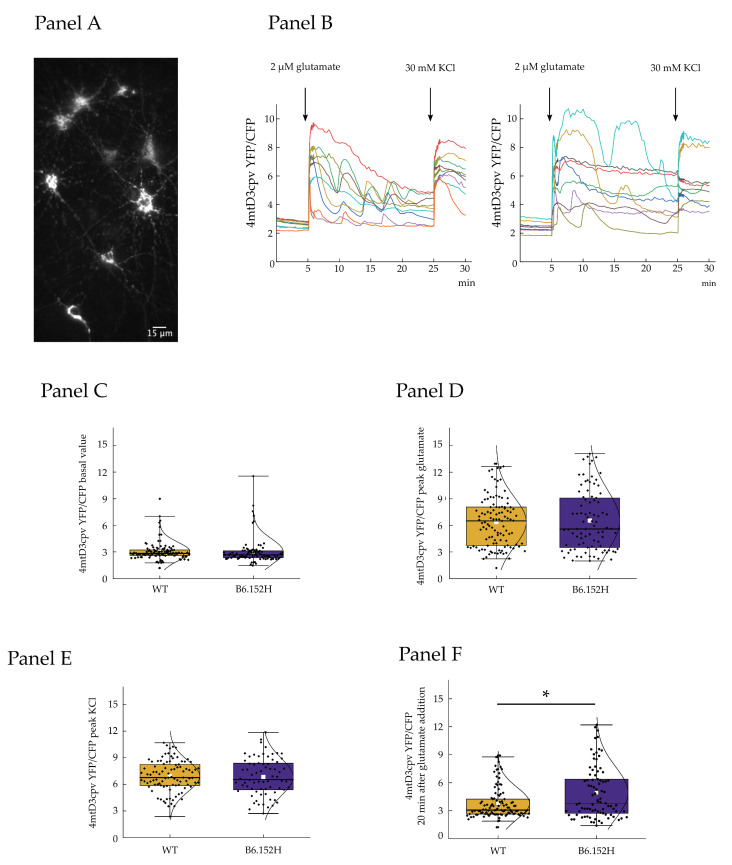
Mitochondrial Ca^2+^ transients in hippocampal primary cultures induced by glutamate. Panel (**A**), representative image of 14 div hippocampal neurons expressing mitochondria-targeted cameleon probe 4mtD3cpv. Panel (**B**), representative traces of mitochondrial Ca^2+^ uptake in hippocampal neurons expressing 4mtD3cpv, left WT; right B6.152H cells. Where indicated the cells were challenged with 2 μM glutamate and 30 mM KCl addition. Panels (**C**–**E**), quantifications of 4mtD3cpv ratio fluorescence for, respectively: C basal values (WT *n* = 111 cells, median = 2.83; B6.152H *n* = 89 cells, median = 2.66, Mann–Whitney probability test, exact prob. = 0.063: at the 0.05 level, the two distributions are not significantly different); (**D**), peak glutamate (WT *n* = 111 cells, median = 6.5; B6.152H *n* = 89 cells, median = 5.6, Mann–Whitney probability test, exact prob. = 0.858: at the 0.05 level, the two distributions are not significantly different); (**E**) peak KCl (WT *n* = 95 cells, median = 6.76; B6.152H *n* = 71 cells, median = 6.54, Mann–Whitney probability test, exact prob. = 0.65: at the 0.05 level, the distributions are not significantly different). Panel (**F**), quantification of the 4mtD3cpv fluorescence ratio measured 20 min after the addition of 2 μM glutamate (WT *n* = 111, median = 3.07; B6.152H *n* = 85 median = 3.79, Mann–Whitney probability test, exact prob. = 0.029: at the 0.05 level, the distributions are significantly different). Box plot: Box height, 25–75%; whiskers, 1–99%; white square, mean; transverse line median; curve, normal distribution equation. The spontaneous movements of mitochondria during the acquisition were controlled for in off-line analysis to ensure the evaluated organelles rested within the boundary of the outlined roi. * *p* ≤ 0.05.

**Figure 9 cells-10-02046-f009:**
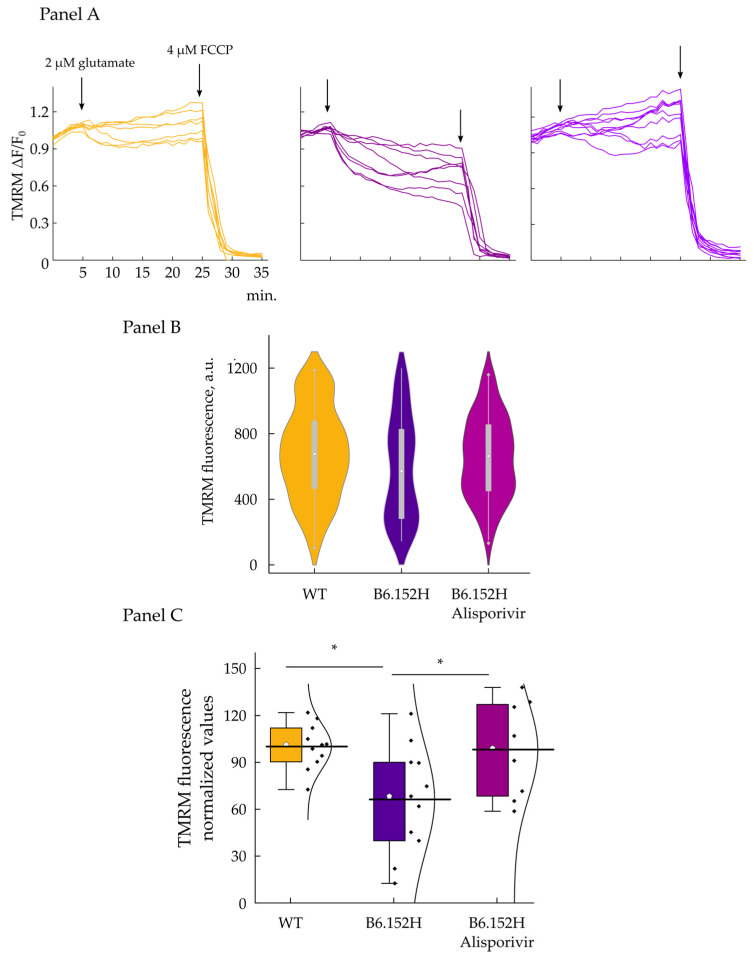
Effect of glutamate treatment on the mitochondrial membrane potential of primary hippocampal neurons. Panel (**A**), representative traces of WT (left), B6.152H (middle), and B6.152H cells treated with 1.5 μM Alisporivir (right), loaded with the potentiometric probe TMRM. Where indicated 2 μM glutamate and 4 μM FCCP were added. Time point TMRM acquisitions are normalized to the difference between the average of initial fluorescence (first 5 min acquisitions), minus the average of fluorescence signal measured after the addition of the uncoupler FCCP (last 3 min acquisitions). Panel (**B**), violin plot, quantification of basal TMRM fluorescence expressed in arbitrary units; respectively for WT, yellow *n* = 167 cells; B6.152H, indigo, *n* = 89 cells; B6.152H plus Alisporivir, purple, *n* = 130 cells. ANOVA one-way test, at the 0.05 level, the population means are not significantly different. Box height, 25–75%; whiskers, 1–99%; white circle, median; transverse line, median. Panel (**C**), quantification of TMRM fluorescence 20 min after 2 μM glutamate addition, expressed as % of initial (mean of 5 acquisitions) fluorescence. Color code as in previous panel. WT, *n* = 11 coverslip (157 cells); B6.152H, *n* = 11 coverslip (87 cells); B6.152H plus 1.5 μM Alisporivir *n* = 8 coverslip (129 cells). ANOVA one-way test: α = 0.05 Tukey and Fisher test, the difference between WT and B6.152H is statistically significant, *p* = 0.02 as well as the difference between B6.152H and B6.152H plus Alisporivir, *p* = 0.048; while the difference between WT and B6.152H plus Alisporivir is not significant. * *p* ≤ 0.05.

**Table 1 cells-10-02046-t001:** Key resources table.

Reagent or Resource	Supplier	Identifier
Chemicals, peptides, cell culture media and supplements		
Poly-D-Lysine	Merck	P6407
Laminin	Merck	L2020
Minimum Essential Medium, MEM	Gibco	32360-026
Basal Medium Eagle, BME	Gibco	41010-026
N2-supplement	Thermo-Fisher	17502048
B27-supplement	Thermo-Fisher	17504044
Carbonyl cyanide 4-(trifluoromethoxy)phenylhydrazone (FCCP)	Merck	C2920
Tetramethylrhodamine, methyl ester (TMRM)	Thermo Fisher	I34361
MitoSOX™ Red	Thermo Fisher	M36008
Cyclosporin H	Merck	SML1575
Bicinchoninic Protein Assay Kit, Quantum Protein	Euroclone	EMP014500
Chemiluminescent reagent Westar Sun	Cyagen	XLS063
Hoechst 33342, Trihydrochloride, Trihydrate	Thermo-Fisher	H1399
MOWIOL^®^ 4-88 Reagent, Poly(vinyl alcohol)	Millipore	475904-M
Antibodies		
Mito profile Total Oxphos rodent WT	Abcam	ab110413
anti HSP 90	BD Bioscience	610418
anti-neurofilament	Merck	N5389
anti glial fibrillary acidic protein	Dako	Z0334
Alexa-Fluor Plus 555 goat anti-mouse IgG	Thermo-Fisher	A32727
Alexa Fluor Plus 488 goat anti-rabbit IgG	Thermo-Fisher	A32731
Recombinant proteins		
jRGECO1	Addgene	100854
4mtD3cpv	Addgene	36324
mito roGFP2-Orp1	Addgene	65001

## Data Availability

Data are reported within the article and in the Appendix A.

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
