# Peer review of "Effects of Mild Excitotoxic Stimulus on Mitochondria Ca2+ Handling in Hippocampal Cultures of a Mouse Model of Alzheimer’s Disease"

_cells, 2021, doi:10.3390/cells10082046_

Round 1
Reviewer 1 Report
The data presented in the cells-1305684 manuscript by Rigotto et al. supports the very careful claims made by the authors. The authors report a decrease in maximal oxygen consumption rate and extracellular acidification rate in primary cortical B6.152H neurons compared to the wildtype. Inhibition of complex I or complex III of the mitochondrial respiratory chain resulted in a faster dissipation of mito membrane potential in B6.152H neurons than wildtype. The authors claim that the mito Ca2+ transients in B6.152H neurons in response to 2 µM glutamate result in sustained higher Ca2+ levels in B6.152H neurons than wildtype. This treatment caused a larger decrease in mito membrane potential of B6.152H neurons than wildtype which could be prevented by inhibiting mito permeability transition pore. These data add to our understanding on Ca2+ handling in hippocampal neurons derived from a mouse model of Alzheimer's disease.
The study is well-conducted as well as well-presented and I have only few minor comments on the study:
- The authors should indicate statistical significance on all the possible figures.
- Please increase the font size of axes labels on figures 3A (right panel), 3C, 9B and 9C.
- There is no mention of the figure 7B 30 mM KCl part in the main text, please include.
- In line 507 the authors state that "These oscillations were more frequently observed in mitochondria of B6.152H cells, ...". However, the authors do not present any data on the quantifications of frequency of Ca2+ oscillations/waves.
- In figure 8, it would be advisable to present the mito Ca2+ transient traces of B6.152H and wildtype neurons next to each other.
- In line 603, the authors do state small effect of alisporivir on TMRM fluorescence in wildtype neurons. This data is important to present.
- The supplementary figure S1 should also be discussed in the results section and please note that this data was not provided to the reviewer.
Author Response
We thank the reviewer to take the time to go through our manuscript and to provide useful comments to it
- The authors should indicate statistical significance on all the possible figures.
We have added statistical significance to the suitable figures
- Please increase the font size of axes labels on figures 3A (right panel), 3C, 9B and 9C.
We have increased the font for the above-mentioned figures
- There is no mention of the figure 7B 30 mM KCl part in the main text, please include.
We have added in the text comment about the significance of KCl addition for figure 7B, please see line 486
- In line 507 the authors state that "These oscillations were more frequently observed in mitochondria of B6.152H cells, ...". However, the authors do not present any data on the quantifications of frequency of Ca2+ oscillations/waves.
We have quantified the percentage of cells in which oscillations could be observed, both WT and B6.152H, please see line 532
- In figure 8, it would be advisable to present the mito Ca2+ transient traces of B6.152H and wildtype neurons next to each other.
We followed the reviewer advice and present mitochondrial Ca2+ transient traces for both WT and B6.152H next to each other to improve the comparison of the different effect exerted by glutamate on the two genotypes
- In line 603, the authors do state small effect of alisporivir on TMRM fluorescence in wildtype neurons. This data is important to present
We have added a supplementary figure S2 showing the effect of Alisporivir on WT neurons treated with glutamate
The supplementary figure S1 should also be discussed in the results section and please note that this data was not provided to the reviewer.
We discussed supplementary figure S1 in the main text, please see line 641 in the Discussion part of the manuscript. We apologize with the reviewer who could not examine the supplementary figure, for convenience, and to avoid loss of documents, we have now provided the supplementary figures at the end of the manuscript.
Reviewer 2 Report
The study evaluated the excitotoxic stimulus effect on mitochondria Ca2+ in hippocampal cultures of a mouse model of Alzheimer's disease. However, few significant issues need to be addressed:
Page 1; Line 14: Abstract lack the objective of the study.
Page 1; Line 20-23: Abstract needs to be constructed in a way that all the outcomes with the significant figures should be given either in the percentage or statistically significant values.
Page 2; Line 71: It would be great if a separate section as "Chemicals" should be included which can show all the chemicals, assay kits used in the study.
Page 2; Line 86: Sentence shouldn't start with numeral value need to change with alphabetical value here and throughout the manuscript.
Page 3; Line 90: Recombinant AAV production should be described in brief even using commercial kit or described in previous research.
Enclosed please check comments in the pdf file.

Author Response
We are grateful to the Reviewer for taking the time to read our manuscript and for providing useful comments to it.
Page 1; Line 14: Abstract lack the objective of the study.
Page 1; Line 20-23: Abstract needs to be constructed in a way that all the outcomes with the significant figures should be given either in the percentage or statistically significant values.
As suggested we have revised the abstract, hopefully making it more centered on the subject of the study, providing clarification about the objective of the work and anticipating the most meaningful and statistically significant results.
Page 2; Line 71: It would be great if a separate section as "Chemicals" should be included which can show all the chemicals, assay kits used in the study.
We have provided a table listing the substances (chemicals, antibodies...) used in the study, please see line 224, we hope this would improve clarity for the interested reader
Page 2; Line 86: Sentence shouldn't start with numeral value need to change with alphabetical value here and throughout the manuscript.
We have corrected the sentence, please see line 94
Page 3; Line 90: Recombinant AAV production should be described in brief even using commercial kit or described in previous research.
A brief, though detailed description of recombinant AAV production has been included, please see line 100